# Obesity Induces an Impaired Placental Antiviral Immune Response in Pregnant Women Infected with Zika Virus

**DOI:** 10.3390/v15020320

**Published:** 2023-01-23

**Authors:** Anna Cláudia Calvielli Castelo Branco, Emily Araujo De Oliveira, Nátalli Zanete Pereira, Ricardo Wesley Alberca, Amaro Nunes Duarte-Neto, Luiz Fernando Ferraz Da Silva, Fernanda Guedes Luiz, Naiura Vieira Pereira, Mirian Nacagami Sotto, Naiara Naiana Dejani, Patrícia Helen Carvalho Rondó, Elyzabeth Avvad-Portari, Zilton Farias Meira De Vasconcelos, Alberto José da Silva Duarte, Tamiris Azamor, Maria Notomi Sato

**Affiliations:** 1Laboratório de Investigação em Dermatologia e Imunodeficiências (LIM56), Faculdade de Medicina, Instituto de Medicina Tropical, Universidade de São Paulo, São Paulo 05403-000, Brazil; 2Departamento de Imunologia, Instituto de Ciências Biomédicas ICB, Universidade de São Paulo, São Paulo 05508-000, Brazil; 3Departamento de Dermatologia, Faculdade de Medicina FMUSP, Universidade de São Paulo, São Paulo 05403-000, Brazil; 4Departamento de Patologia, Faculdade de Medicina FMUSP, Universidade de São Paulo, São Paulo 05403-000, Brazil; 5Instituto Pasteur, São Paulo 01311-000, Brazil; 6Laboratório de Biologia Molecular, Departamento de Fisiologia e Patologia, Centro de Ciências da Saúde, Universidade Federal da Paraíba, João Pessoa 58051-900, Brazil; 7Departamento de Nutrição, Faculdade de Saúde Pública FSP, Universidade de São Paulo, São Paulo 01246-904, Brazil; 8Instituto Fernandes Figueira, Fundação Oswaldo Cruz FIOCRUZ, Rio de Janeiro 22250-020, Brazil; 9Instituto de Tecnologia em Imunobiológicos, Bio-Manguinhos, Fundação Oswaldo Cruz FIOCRUZ, Rio de Janeiro 21040-900, Brazil

**Keywords:** obesity, pregnancy, placenta, Zika virus, antiviral factors

## Abstract

Obesity is increasing in incidence worldwide, especially in women, which can affect the outcome of pregnancy. During this period, viral infections represent a risk to the mother, the placental unit, and the fetus. The Zika virus (ZIKV) outbreak in Brazil has been the cause of congenital Zika syndrome (CZS), with devastating consequences such as microcephaly in newborns. Herein, we analyzed the impact of maternal overweight/obesity on the antiviral factors’ expression in the placental tissue of Zika-infected mothers. We accessed placentas from women with and without obesity from 34 public health units (São Paulo) and from Zika-infected mothers with and without obesity from the Clinical Cohort Study of ZIKV pregnant women (Rio de Janeiro, Brazil). We first verified that obesity, without infection, did not alter the constitutive transcriptional expression of antiviral factors or IFN type I/III expression. Interestingly, obesity, when associated with ZIKV infection, showed a decreased transcriptional expression of RIG-I and IFIH1 (MDA-5 protein precursor gene). At the protein level, we also verified a decreased RIG-I and IRF-3 expression in the decidual placenta from the Zika-infected obese group, regardless of microcephaly. This finding shows, for the first time, that obesity associated with ZIKV infection leads to an impaired type I IFN downstream signaling pathway in the maternal–fetal interface.

## 1. Introduction

The number of overweight people has tripled since 1975 worldwide, counting more than 1.9 billion in 2016, of which more than 650 million are obese [1]. In Brazil, in 2019, approximately 27% of the population was obese (Body Mass Index; BMI ≥ 30), and 60% were overweight (BMI ≥ 25) [2]. Obesity is characterized by the accumulation of white adipose tissue, which may be related to disorders such as insulin resistance, diabetes, and, consequently, metabolic syndrome [3]. It also may alter the immune system [4], starting with the activation of innate immunity receptors, such as the Toll-like receptor 4 (TLR4), in addition to lipopolysaccharide (LPS), by circulating saturated fatty acids (SFAs) in obese individuals. Moreover, signaling via TLR4 induces nuclear factor, such as IRF3 and NF-κB, translocation, leading to the production of cytokines, the activation of the cyclooxygenase-2 (COX2) pathway, and consequently, chronic inflammation in people with obesity [5]. Between the sexes, women with obesity have higher levels of peripheral body fat compared to men with obesity [6]. Also, obesity represents an increased risk for pregnancy complications, such as gestational hypertension, preeclampsia, gestational diabetes mellitus, and a higher incidence of congenital defects [7]. Other later complications emerge in gestation, delivery, or postpartum [8]. Globally, in 2014, 38.9 million overweight and obese pregnant women were estimated [9]. Specifically in Brazil, in 2018, almost 30% of prepregnant women were overweight, and almost 20% obese [10].

Obesity may also increase the risk of severe infectious diseases [4], such as dengue [11], influenza [12], and more recently COVID-19 [13]. Maternal obesity decreased the odds of anemia but increased the odds of developing cardiovascular diseases, such as hypertensive disorders, in HIV-infected pregnant women [14]. The influence of obesity during pregnancy is still unknown in association with infections such as the Zika virus (ZIKV) and the risk of congenital malformation.

The ZIKV outbreak in Brazil began in 2015 but exhibited its peak of occurrence in 2016 when 216,201 cases were registered [15]. In 2017, Brazil had around 13,490 reported cases and 2653 confirmed microcephaly cases caused by ZIKV. The Brazilian infection by this virus was considered the most serious reported in Latin America, due to the association of the infection during pregnancy with microcephaly in newborns [16,17], in addition to other brain defects, named congenital Zika syndrome (CZS). Regarding the epidemiological situation of CZS in Brazil, between 2015 and 2020, 3559 cases were confirmed (18.4% of the total cases reported), with 10 births and one miscarriage confirmed in 2020. Although the emergency period had ended, the number of microcephaly cases among newborns in Brazil in 2021 amounted to 287 children, showing that the virus is still circulating, and transmission persists at low levels [18].

Some risk factors specifically influence the incidence of the transmission to the fetus: the timing of the infection in pregnancy, the order of the infection, whether primary, reinfection, or chronic, the time of membrane rupture, and the type of delivery.

Zika virus is a single-stranded RNA virus from the *Flaviviridae* family [19]. As an immune evasion strategy, nonstructural ZIKV proteins, including NS1, NS4A, and NS4B, can inhibit the activation of cytosolic RIG-I and MAVS receptors [20], impairing the IFN levels type I. IFNs are the main cytokines that mediate the antiviral response, as they induce interferon-stimulated genes (ISGs) that act to decrease viral replication. Another evasion mechanism is the degradation of STAT2, through binding with the ZIKV NS5 protein [21]. CZS has been associated with the IFN Alpha Receptor 1, which contributes to exacerbating the levels of type I IFN with insufficient type III IFN in placenta at term [22]. ZIKV infection is able of morphologically altering the placenta, as hyperplasia of Hofbauer macrophages in the villi [23,24], deciduitis, infiltration of the lymphocytes associated with vasculitis, and increased levels of the apoptosis inhibitor, Bcl-2, in syncytiotrophoblast. These parameters promote the persistence of ZIKV in the placenta, facilitating its replication and fetal transmission [25].

Herein, for the first time, we analyzed the impact of maternal overweight/obesity on the antiviral factors’ expression in the placenta of Zika-infected women during pregnancy. The data reinforce that obesity associated with ZIKV infection leads to an impaired type I IFN signaling pathway in the maternal–fetal interface.

## 2. Materials and Methods

### 2.1. Human Subjects

This study enrolled pregnant women with and without overweight/obesity, divided into groups of non-infected and Zika-infected, to access placental samples at the time of delivery.

In total, 39 non-infected pregnant women, based on the antenatal body mass index (BMI), were determined not to have obesity (BMI 18.5–24.9; *n* = 11) and to have overweight/obesity (BMI 25–39.9; *n* = 28). The women were selected from 34 public health units in Araraquara city, São Paulo, from a Cohort Study, coordinated by Prof Patricia Rondó, University of São Paulo. An informed consent form approved by the Ethics Committee, number: 59787216.2.0000.5421, was collected from the women. Samples of normal delivery or cesarean section, from women older than 18 years and with a gestational age above 37 weeks, were included. The exclusion criteria were positive serology for syphilis, HIV-1, hepatitis B, C, and toxoplasmosis, as well as clinical or obstetric complications associated with increased postoperative complications. Placentas and samples from preterm and post-term deliveries were also excluded.

In total, 30 Zika-infected women, enrolled in the Clinical Cohort Study of ZIKV pregnant women and their infants at a maternal and child hospital (Instituto Nacional de Saúde da Mulher, da Criança e do Adolescente Fernandes Figueira, Fiocruz) in Rio de Janeiro, Brazil, collected in the period of 2015–2016, (IRB/CAAE: 52675616.0.000.5269) were included in the study. Congenital infection by ZIKV was confirmed during pregnancy by the positive PCR of urine, blood, or placenta samples. We accessed placentas from the women without obesity (BMI 18.5–24.9; *n *= 14) and with overweight/obesity (BMI 25–39.9; *n *= 16). The samples from the mothers were tested and excluded for HIV, evidence of past Dengue virus, and Chikungunya virus (CHIKV) infection. This cohort included pregnant adult women >18 years of age, and the exclusion criteria included maternal HIV infection and pregnancies complicated by other congenital infections, known to cause infant neurologic damage (e.g., TORCH, CHIKV). Placentas and samples from preterm and post-term deliveries were also excluded. The placental samples were collected at the time of delivery from the umbilical cord insertion region. The clinical data of infants who had an adverse neurologic outcome at the time of birth, such as microcephaly (head-circumference z score of less than −2), were used in this study.

### 2.2. Gene Expression by Real-Time PCR

The real-time polymerase chain reaction (PCR) for antiviral factors was performed in the decidua and chorionic villus from the placenta samples, from non-infected and Zika-infected women, both with and without obesity. From the ZIKV-infected women, we used the repository data, previously published by Azamor et. al. 2021 [22], to perform new biological analysis. The placental specimens from non-infected placenta were cryopreserved and processed using a Tissue Ruptor (Qiagen, Hilden, Germany). Total RNA was extracted from the placental decidua and villus using the RNeasy Plus Mini Kit (Qiagen). The RNA levels were measured in a NanoDrop ND-1000 spectrophotometer (Thermo Fisher Scientific, Waltham, MA, USA). The reverse transcription of the total RNA from each sample was performed using a Reverse Transcriptase Kit, according to the manufacturer’s instructions (Bio-Rad, Berkeley, CA, USA). The real-time PCR amplification reaction was conducted in the Applied Biosystems 7500 Real-Time PCR Systems (Thermo Fisher Scientific), using 5 ng/µL of cDNA and 1 µM of specific primers, with the Applied Biosystems SYBR Green PCR Master Mix (Thermo Fisher Scientific). The primers for all target genes (TLR3, TLR7, IRF-3, STING, RIG-1, MxA, ISG-15, IFN-α, IFN-β and IFN-λ), using YWHAZ as an internal control, are listed in Appendix A. Data analysis was performed with the 7500 Software v2.0.6 (Applied Biosystems, Waltham, MA, USA) according to the delta-CT method [26].

### 2.3. Morphological Analysis of the Placenta

Histological sections of 4µm thickness were taken from biopsies of paraffin-embedded placentas on silanized slides (Sigma Chemical Co., St. Louis, MO, USA). The histological sections were deparaffinized in xylene baths and subsequently hydrated in decreasing concentrations of ethanol. The slides were stained with hematoxylin–eosin and analyzed under a microscope. The histopathological parameters comparing the placental tissue slides of the mothers ZIKV-infected with or without obesity groups were evaluated by the pathologist.

### 2.4. Antiviral Factors’ Expression by Immunohistochemistry

All tissue specimens were formalin-fixed and embedded in paraffin, and 4-µm thick sections were obtained on silanized glass slides. The slides were dewaxed in xylene and hydrated through a graded series of ethanol. Endogenous peroxidase was blocked with 3% hydrogen peroxide. Antigen retrieval for IFNA4, IRF3, and RIG-I analysis was performed by incubation of the slides in 50 mM Tris-ethylenediaminetetraacetic acid (EDTA) buffer, pH 9.0 (code 52367, Dako, Carpinteria, CA, USA), in a water bath for 20 min at 95 °C, followed by washing in running water and distilled water for 5 min each. Then, they were incubated overnight at 4 °C in the presence of the primary antibody, polyclonal rabbit anti-IFNA4 primary antibody (1:300, ab232899, Abcam, Boston, MA, USA), monoclonal rabbit anti-IRF3 (1:60, ab76409, Abcam), or polyclonal goat anti-RIG-I/DDX58 (1:30, ab111037, Abcam). The specific antigen–antibody reaction for IFNA4 was detected with the NovoLink Polymer Detection System Kit (code RE7280-CE, Leica Microsystems Inc., Newcastle Upon Tyne, UK), IRF3 was detected with the Detection Kit-Micro-polymer (ab236466, Abcam, Boston, MA, USA), and RIG-I/DDX8 was detected with the Stain MAX PO Universal Immuno-peroxidase Polymer (Histofine 414161F, Nichirei Biosciences Inc., Tokyo, JPN), according to the manufacturer’s instructions. The reactions were visualized using the 3,3′-diaminobenzidine-tetrahydrochloride (DAB) chromogen (Sigma) and counterstained with Carazzi hematoxylin. All reactions were accompanied by both positive (placenta and tonsil tissue) and negative controls. For the negative control of nonspecific binding, the primary antibodies were replaced with normal serum, under the same experimental conditions. For each protein analysis, we used five random regions on the slides for the placental villus and three regions for the decidua. The Image-Pro plus version 4.5.0.29 software was used for quantification, by counting the positive area or optical density intensity (ODI) by the total area or length of the basement membrane, for the placental decidua. For normalization, a 10× ocular lens was used for the decidua and 20× for the placental villus.

### 2.5. Statistical Analyses

Comparisons between the obese and non-obese samples were performed with the Mann–Whitney U test. The level of significance considered was *p* ≤ 0.05.

## 3. Results

### 3.1. Gestational Obesity and the Profile of Antiviral Immune Transcripts in the Placental Tissue 

To access the influence of gestational obesity on the placental antiviral response, we enrolled placenta from women with overweight/obesity (*n *= 28) and women without obesity (*n *= 11) (Figure 1). The women with obesity showed a higher placental weight (Figure 1B) compared to the women without obesity (*p *= 0.0028). Moreover, the maternal body fat was increased in the women with obesity compared to the control group (*p* < 0.0001). The type of delivery was similar between vaginal birth and C-section (Figure 1A).

We first assessed the placental tissue of the women with and without obesity, who were full-term parturient, shortly after delivery, to verify whether the pregestational obesity altered the immunological factors associated with the antiviral response (Appendix A). The expression values were normalized with the internal YWHAZ gene used in placentas, due to its stable expression in this tissue [27].

A similar profile of the constitutive transcriptional expression of antiviral factors, STING, RIG-I, MxA, and ISG15 was observed between the non-obese and obese groups (Figure 2A). Other factors such as the innate viral recognition receptors, TLR3 and TLR7, the interferon regulatory factor 3 (IRF3) (Figure 2B), as well as type I (IFNα and β) and III IFNs (IFNλ, Figure 2C) did not change the expressions between the groups. These data show that despite the BMI of the women in this cohort, there was an important regulation of the antiviral/inflammatory factors to maintain local homeostasis. 

### 3.2. Decreased Antiviral Immune Factors Expression in Placenta due to Gestational Obesity in ZIKV Infection

Obesity, as a metabolic syndrome, may significantly alter the immune response, leading to a chronic inflammation status [4]. It is not clear whether inflammation triggered by obesity during pregnancy may influence antiviral immunity, as a risk factor for infections. Therefore, we assessed the cohort of women with ZIKV during the outbreak that occurred in Rio de Janeiro between 2015–2016, from the Fernandes Figueira Institute/Fiocruz in Rio de Janeiro/Brazil. Demographic data from women with and without overweight/obesity in the cohort of ZIKV-infected mothers are shown in Figure 3. 

The gestational age of full-term pregnancy was similar between the mothers with and without obesity (Figure 3C; *p *= 0.6563), and despite no statistical difference, the non-obese newborn weight median was trend higher than the obese group. The time of ZIKV infection during pregnancy occurred mostly in the first trimester (Figure 3B; 58.3%), in the women without obesity, differently that what was observed for women with obesity, when the ZIKV infection occurred mostly in the second trimester of gestational period (50%), followed by the first trimester (31.2%). The type of delivery of ZIKV-infected women with obesity was mainly by C-section (Figure 3A; 91.7%) compared to the ZIKV+-women without obesity (45.5%). For the clinical aspect of the newborns, detectable at the time of delivery were microcephaly and eye abnormalities, considered important characteristics of CZS. Microcephaly was detected in 6/11 cases of the ZIKV+-non-obese group (Figure 3D) and 1/12 cases from the ZIKV+-obese group, probably due to the time of the ZIKV infection. Similar cases of eye abnormalities were observed in both groups. In the histopathological analysis (Table 1), we observed a more intense (3+) trophoblast necrosis in 45.4% of the placentas from the mothers with obesity in contrast to the moderate level of 87.5% (2+) in the mothers without obesity. Other findings were at similar intensity in both infected mothers’ groups, such as villitis, inter-villositis, thrombi, calcification, and so forth.

To assess the antiviral transcriptional profile, we used a database published by Azamor et al. in 2021 [22], from the same cohort of ZIKV-infected pregnant women. Interestingly, we observed a decreased expression of RIG-I and IFIH1, the MDA-5 protein precursor gene, in the placental tissue of the mothers with obesity (Figure 4A; *p *= 0.0249; *p *= 0.0108, respectively), compared to the non-obese group, during ZIKV infection.

ZIKV is composed of single-stranded RNA and can be recognized by TLRs. Expressions of TLR3 and TLR7 were similar between the groups; however, the downstream signaling pathway, IRF3, was downregulated in the placentas of the infected mothers with obesity (Figure 4B *p *= 0.036).

The type I and III IFNs expressions did not change (Figure 4C) according to the maternal BMI. It is possible that as we assessed full-term placental tissue at the time of delivery, some differences could no longer be observed, since the infection may have occurred early in the pregnancy.

### 3.3. The IRF-3/RIG-1 Axis Is Altered in Pregnant Women with Obesity in ZIKV Infection

Next, we aimed to access the protein expression of some target factors observed at the transcriptional level. The maternal placental side, decidua, and the fetal side, villus, from ZIKV-infected mothers with and without obesity were analyzed by immunohistochemistry for IRF-3, RIG-I, and IFN-α. The positive area was measured, and optical density intensity (ODI) was performed for each of the markers. 

Figure 5 shows a decreased RIG-I protein expression level in the decidua of the ZIKV-infected mothers with obesity compared to the non-obese group, both in the positive area (*p *= 0.001) evaluation and the ODI (*p *= 0.001). This finding was like that obtained from the transcript expression level. We also verified a decreased expression of IRF-3 in the placental decidua of mothers with obesity (Figure 6; area *p *= 0.0015; ODI *p *= 0.0015; area/basement membrane *p *= 0.0004). These data reinforce that metabolic syndrome has a direct impact on the antiviral response of the maternal–fetal interface.

The expression of IFN-α was similar between the groups (Figure 7). IFN-β or type-III IFN were not checked. The downregulation of the RIG and IRF-3 was not related to the placentas of the mothers that had newborns with microcephaly, whereas it was related to the obesity associated with ZIKV infection.

## 4. Discussion

Obesity has an impact on pregnancy leading to severe alterations at the maternal–fetal interface and in the fetus, among them, a deficit in antiviral immunity. Our findings showed, for the first time, an impaired RIG-I/IRF3 axis in the decidua of the placenta from mothers with obesity in association with ZIKV infection.

Pregestational obesity, as expected, showed an increased percentage of fat, increased weight of the placentas, and normal weight of newborns at birth, both in the ZIKV infected and non-infected cohorts. However, maternal obesity is usually correlated with fetal macrosomia (≥4000 g at birth), as major obstetric adversity [28], this difference was probably due to our inclusion criteria of full-term pregnancy (between 37 and 42 weeks), without obstetric complications or preterm births, to avoid the alteration in the placenta’s immunologic profile. 

When assessing the placental expression of factors related to viral recognition (TLR3, TLR7, and IRF3) and antiviral response (IFNs I and III, STING, RIG-I, MxA, and ISG15) from parturient with pregestational obesity (BMI ≥ 25), no changes were detected between the obese and non-obese samples. These results showed that despite obesity, the decidua revealed a homeostatic control, at transcriptional antiviral factors, in the innate immune response against viruses. Our limitation was that, in these cases, protein analysis was not performed. On the other hand, it is important to emphasize that the gestational environment is naturally programmed to be tolerogenic, preventing exacerbated immune responses and ensuring a controlled environment in effector responses and inflammatory reactions [29]. It has already been described that the impaired IgG immune response to Merkel cell polyomavirus (MCPyV) during pregnancy can favor viral replication and the occurrence of spontaneous abortions [30].

Obesity in pregnancy produces the abnormal production of hormones such as adipokines, TNF, and IFNs, as well as the circulating short-chain saturated fatty acids, that activate the immune system [31], which may compromise the effector function of antiviral mechanisms. In the COVID-19 pandemic in Brazil, as well as in other countries, obesity was considered a greater gestational complication with maternal mortality, causing 12% of maternal deaths from January to November 2020 [32].

Since Brazil was the country with the most serious Zika virus infection cases, leading to CZS cases, in partnership with researchers from Fundação Oswaldo Cruz, Rio de Janeiro, Brazil, we had access to a cohort of pregnant women with congenital ZIKV infection from 2015 to 2016. We analyzed the real-time PCR database already published by the group [22] to compare the antiviral factors’ profile between placentas from ZIKV-infected mothers with and without obesity. An additional limitation in our study was the sample size, due to the rarity of the samples, and because we needed to exclude the placentas of pre- and post-term pregnant women, with other infections, and obstetric complications, in order to increase the accuracy of the investigation.

Interestingly, only one case of microcephaly was found in the obesity group, whereas six cases were founded in the non-obese group. This showed that most case of microcephaly occurred when infection occurred mainly in the first trimester, as verified in 60% of the ZIKV infection in non-obese cohort. It is already well known that the timing of ZIKV infection in the first trimester is associated with the occurrence of fetal microcephaly [33].

ZIKV infection triggered several histopathological alterations in the placenta from both mother groups, such as increased trophoblastic necrosis, fibrinoid necrosis of the villi, calcification, decidual inflammation, and hyperplasia of Hofbauer macrophages. This alteration was unrelated to the occurrence of fetal microcephaly. Zika virus infection during pregnancy has already been shown to cause several changes in the placenta [25], mainly macrophage hyperplasia [23], which may indicate a replicative niche in this microenvironment.

It should be noted that ZIKV infection in association with obesity, induced a decrease in the transcriptional RIG-I, IFIH1, and IRF3 expression in the placental tissue, compared to the non-obese group. This was also confirmed by the decrease in RIG-I and IRF-3 protein expression at the decidual portion of the placenta. Although obesity does not alter these factors’ expression constitutively, ZIKA invasion triggered a dysfunctional antiviral response in the maternal–fetal interface. Moreover, the maternal decidua has a wide variety of immune cells in its composition and represents an important barrier in the fight against infectious agents during the gestational period [34]. It is important to point out that the differences observed in the period of congenital infection between the non-obese and obese groups is a limitation of the study.

The IRF3 and RIG-I axis plays an important role during ZIKV infection, as one of the main sensors for this virus [35]. Importantly, one of the mechanisms through which ZIKV can antagonize the interferon response is through interaction with the RIG-I-MAVS [36]. Moreover, regarding obesity, it has been described that the mice deficient in the RIG-I receptor had worsening weight gain, white adipose tissue, and insulin resistance, induced by a hyperlipidemic diet [37].

Taken together, we observed that pregestational obesity in association with ZIKV infection altered the RIG-I/IRF3 axis in the placenta. This impaired immune antiviral response in gestational obesity, together with the viral strategies to evade host immunity, is a risk factor for the fetus.

## Figures and Tables

**Figure 1 viruses-15-00320-f001:**
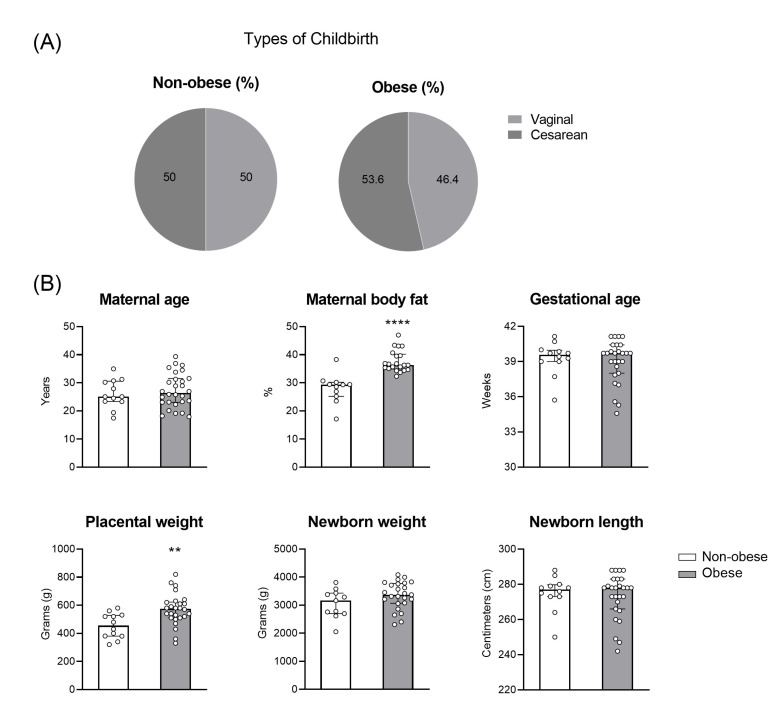
Demographic data of non-obese and obese mothers and newborns. A cohort of full-term pregnant women were assessed from the non-obese (white bar; *n *= 11) and obese (gray bar; *n *= 28) groups. The type of delivery (**A**), maternal age, maternal body fat, gestational age, placental and newborn weight, and newborn length (**B**) was obtained from the maternity hospitals. The bars represent the median and interquartile range of the values. ** *p* ≤ 0.01; **** *p* ≤ 0.0001.

**Figure 2 viruses-15-00320-f002:**
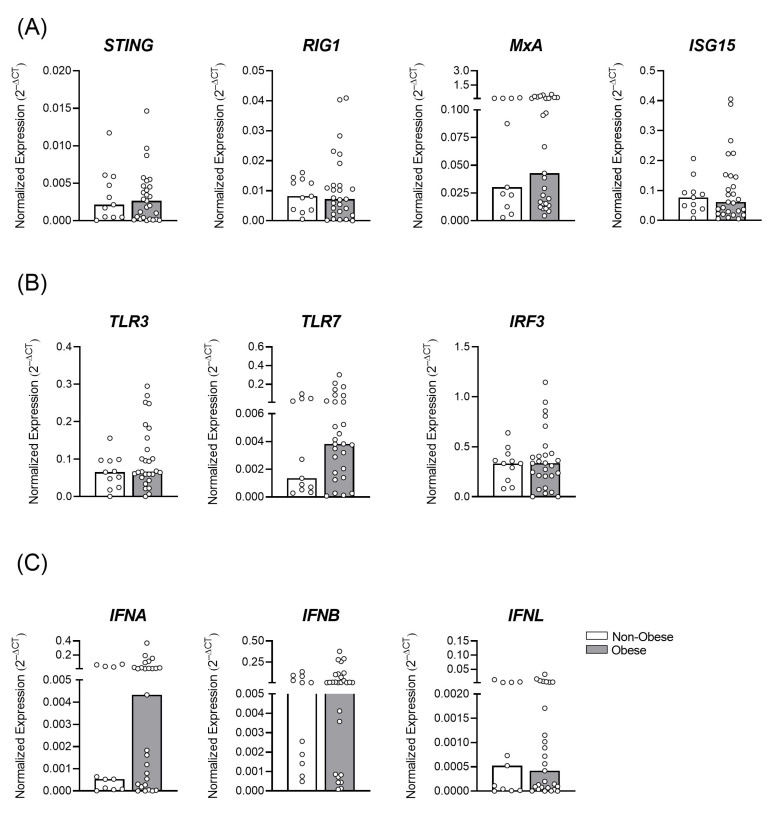
Placental tissue from mothers with and without obesity shows a similar profile of transcriptional antiviral factors’ expression. Placentas, collected after delivery, of full-term pregnant women, were assessed from the non-obese (white bar; *n *= 11) and obese (gray bar; *n *= 28) groups. The mRNAs of the antiviral factors (**A**) STING, RIG1, MxA, and ISG15, (**B**) Toll-like receptors and the intracellular pathway of activation, TLR3, TLR7, and IRF3, and (**C**) interferons, IFNA, IFNB, and IFNL, were evaluated by RT qPCR. The bars represent the median and interquartile range of the values.

**Figure 3 viruses-15-00320-f003:**
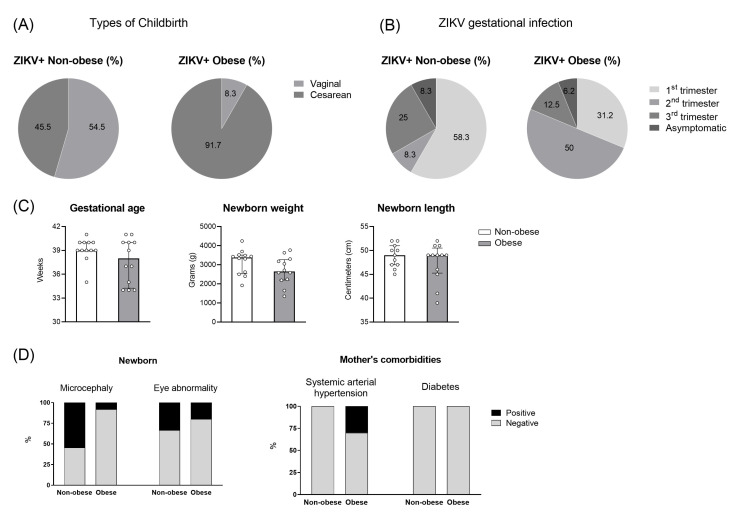
Demographic data of non-obese and obese mothers and newborns with Zika virus (ZIKV) infection. A cohort of full-term pregnant women were assessed from the non-obese (white bar; *n *= 14) and obese (gray bar; *n *= 16) ZIKV+ congenital infection groups. The type of delivery (**A**), gestational trimester of ZIKV infection (**B**), gestational age, newborn weight, and length (**C**), and newborn’s abnormalities and mother’s comorbidities (**D**) was obtained from the maternity hospitals. The bars represent the median and interquartile range of the values.

**Figure 4 viruses-15-00320-f004:**
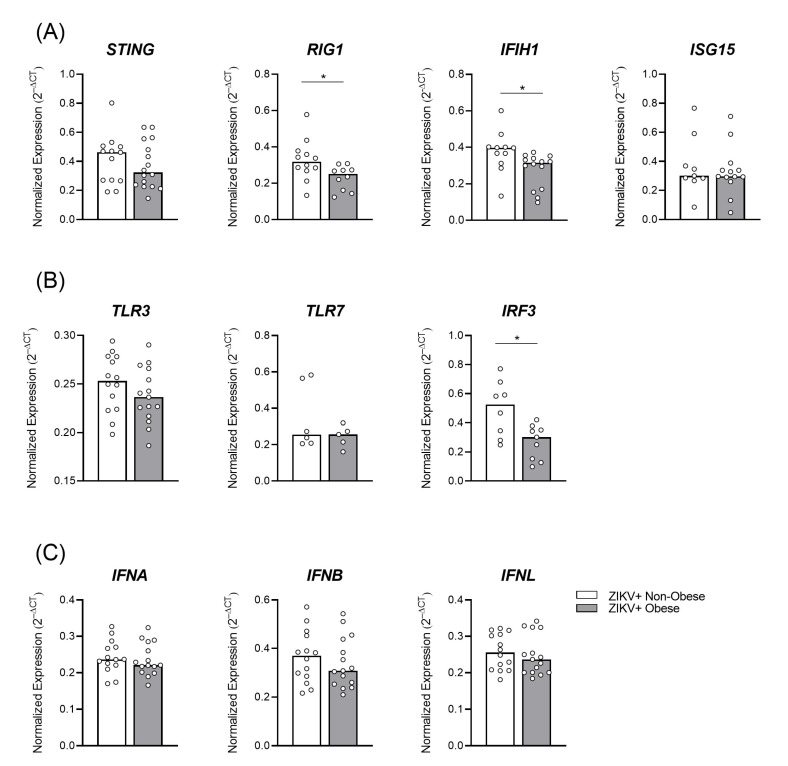
Altered placental antiviral factors’ expression triggered by ZIKV infection in gestational obesity. Placentas, collected after delivery, from ZIKV+ infected mothers were separated into the non-obese (white bar; *n *= 14) and obese (gray bar; *n *= 16) groups. Placental cotyledon fragments were isolated with the fetal and maternal parts. The mRNAs of antiviral factors, (**A**) STING, RIG1, MxA, and ISG15, (**B**) TLR3, TLR7, and IRF3, and (**C**) IFNA, IFNB, and IFNL were evaluated by RT qPCR, from database published by Azamor et al., 2021 [22]. The bars represent the median and interquartile range, * *p* ≤ 0.05.

**Figure 5 viruses-15-00320-f005:**
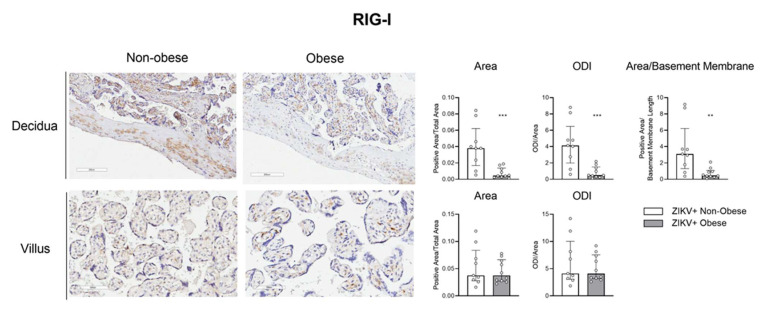
Decreased RIG-I protein level in the placental decidua of ZIKV-infected mothers with obesity. Placentas, collected after delivery, from ZIKV+ infected mothers were separated into the non-obese (white bar; *n *= 9) and obese (gray bar; *n *= 10) groups. Placental cotyledon fragments were isolated with the fetal and maternal parts. RIG-I protein expression was analyzed in the decidua and the placental chorionic villi by immunohistochemistry, representing an average of three distinct regions of each slide for the decidua (10×) and five for the villi (20×). The bars represent the median and interquartile values for the positive area/total area, the positive ODI/total area for the decidua and villus, and the positive area/basement membrane length for the decidua, ** *p* ≤ 0.01, *** *p* ≤ 0.001.

**Figure 6 viruses-15-00320-f006:**
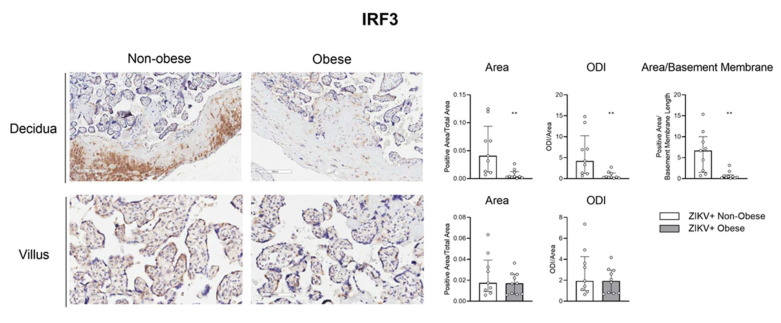
Decreased IRF-3 protein expression level in the placental decidua of ZIKV-infected mothers with obesity. Placentas, collected after delivery, from ZIKV+ infected mothers were separated into the non-obese (white bar; *n *= 9) and obese (gray bar; *n *= 10) groups. Placental cotyledon fragments were isolated with the fetal and maternal parts. IRF-3 protein expression was analyzed in the decidua and placental chorionic villi by immunohistochemistry, representing an average of three distinct regions of each slide for the decidua (10×) and five for the villi (20×). The bars represent the median and interquartile values for the positive area/total area, the positive ODI/total area for the decidua and villus, and the positive area/basement membrane length for the decidua, ** *p* ≤ 0.01.

**Figure 7 viruses-15-00320-f007:**
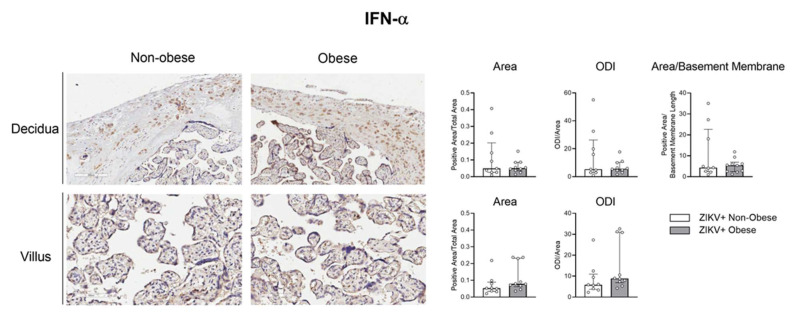
IFN-α expression in the placenta of ZIKV-infected mothers with obesity. Placentas, collected after delivery, from ZIKV+ infected mothers were separated into the non-obese (white bar; *n *= 9) and obese (gray bar; *n *= 10) groups. Placental cotyledon fragments were isolated with the fetal and maternal parts. IFN-α protein expression was analyzed in the decidua and placental chorionic villi by immunohistochemistry, representing an average of three distinct regions of each slide for the decidua (10×) and five for the villi (20×). The bars represent the median and interquartile values for the positive area/total area, the positive ODI/total area for the decidua and villus, and the positive area/basement membrane length for the decidua.

**Table 1 viruses-15-00320-t001:** Histological characteristics of placentas from non-obese and obese mothers with Zika virus (ZIKV)congenital infection.

Characteristic	ZIKV+ Non-Obese (*n* = 8)			ZIKV+ Obese (*n* = 11)		
	Score	n/N	%	Score	n/N	%
Trophoplast necrosis	+	(0/8)	0	+	(2/11)	18.2
	+ +	(7/8)	87.5	+ +	(4/11)	36.4
	+ + +	(1/8)	12.5	+ + +	(5/11)	45.4
Villitis	-	(7/8)	87.5	-	(8/11)	72.7
	+	(1/8)	12.5	+	(3/11)	27.3
Intervillositis	-	(8/8)	100	-	(10/11)	90.9
	+	(0/8)	0	+	(1/11)	9.1
Intervillous thrombi	-	(2/8)	25	-	(0/11)	0
	+	(1/8)	12.5	+	4/11)	36.4
	+ +	(4/8)	50	+ +	(6/11)	54.5
	+ + +	(1/8)	12.5	+ + +	(1/11)	9.1
Hofbauer cell hyperplasia	+	(0/8)	0	+	(1/11)	9.1
	+ +	(6/8)	75	+ +	(8/11)	72.7
	+ + +	(2/8)	25	+ + +	(2/11)	18.2
Fibrinoid necrosis of villi	-	(0/8)	0	-	(2/11)	18.2
	+	(2/8)	25	+	(2/11)	18.2
	+ +	(6/8)	75	+ +	(7/11)	63.6
Villi Calcification	-	(0/8)	0	-	(2/11)	18.2
	+	(5/8)	62.5	+	(3/11)	27.3
	+ +	(0/8)	0	+ +	(3/11)	27.3
	+ + +	(3/8)	37.5	+ + +	(3/11)	27.3
Fetal surface changes	-	(8/8)	100	-	(11/11)	100
Syncytial knots	+ +	(2/8)	25	+ +	(6/11)	54.5
	+ + +	(6/8)	75	+ + +	(5/11)	45.4
Decidual inflammation	-	(6/8)	75	-	(7/11)	63.6
	+	(2/8)	25	+	(4/11)	36.4

- non-present; + low level; + + moderate level; + + + intense level.

## Data Availability

No new data were created or analyzed in this study. Data sharing is not applicable to this article.

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
