# Peer review of "Obesity Induces an Impaired Placental Antiviral Immune Response in Pregnant Women Infected with Zika Virus"

_viruses, 2023, doi:10.3390/v15020320_

Round 1

Reviewer 1 Report

1. This is well written study with comprehensive information, showing some evidence that obesity induces an impaired placental antiviral immune response in pregnant women infected with Zika virus.

2. There were clinical differences bewteen obese and non-obese patients e.g., the mean of newborn weight of obese 2650g was much lower than non-obese 3385g which is very unusual (should be reversed). Can this be explained and could this affect the conclusion?

3. There were differences in timing of infection between obese and non- obese patients. 7 of 12 of non-obese were infected in first trimester while 8 of 16 of the obese group were infected in second trimester. Could these have accounted for the differences in results? 

Author Response

Reply to the Review Report (Reviewer 1)

1. This is well written study with comprehensive information, showing some evidence that obesity induces an impaired placental antiviral immune response in pregnant women infected with Zika virus.

We appreciate the reviewer’s comments. We have addressed all the comments bellow, and the modifications in the manuscript can be tracked in color yellow.

2. There were clinical differences between obese and non-obese patients e.g., the mean of newborn weight of obese 2650g was much lower than non-obese 3385g which is very unusual (should be reversed). Can this be explained, and could this affect the conclusion?

Although the differences found in the weight of newborns were not statistically different (p value 0.1239), it is possible that the small difference between the mean gestational age of the groups (39 weeks non-obese x 38 weeks obese) as well as for the period of congenital infection (mostly in the 1st trimester in the non-obese group and 2nd trimester in the obese group) may be related to the observed newborn weight. Considering that is a heterogeneous value, this do not seems to affect our conclusion.

We appreciate for pointing this out and have added this information both in the results (lines 238-244) and in the manuscript discussion (lines 327-333).

3. There were differences in timing of infection between obese and non- obese patients. 7 of 12 of non-obese were infected in first trimester while 8 of 16 of the obese group were infected in second trimester. Could these have accounted for the differences in results?

Yes, we agree with the Reviewer thinking. The occurrence of ZIKV infection being mostly in the 1st trimester in the non-obese group, and in the 2nd trimester in the obese group may be one of the causes for the observed difference in the occurrence of fetal microcephaly (54.5% in the non-obese group and 8, 3% in the obese group), and we believe that it may also affect the protein composition of the antiviral response, being a limitation of the study.

The composition of the placental immune system is directly related to the gestational trimester. To minimize these differences, and allow data with fewer variables, we excluded all placental samples from other gestational ages (pre and post term).

This was added in the casuistic (lines 128-129), results (lines 238-244), and in the manuscript discussion (lines 380-382).

Reviewer 2 Report

The work is fine and well conducted. The authors investigated the impact of maternal overweight/obesity on the antiviral factors’ expression in the placental tissue of Zika-infected mothers. Results indicate that obesity associated with ZIKV infection leads to an impaired type I IFN downstream signaling pathway in the maternal–fetal interface.
1.    Obesity rates among pregnant females worldwide and specifically in brasil should b included in the introduction (PMID: 30092099 and PMID: 34258876)
2.    This sentence in lines 56-60 is unclear and should be rephrased or subdivided in two sentences.
3.    Figures 1, 2 and 3 words should be enlarged as being difficult to read
4.    Tables 1, 2 and 3 are actually figures. Please replace these figures with text
5.    Pregnant females might become prone to viral infection as a consequence of the physiological immune modulation process occurring during the gestation period (PMID: 34970247, PMID: 33133091). Please include this notion
6.    Please include a supporting reference for the statement present  in line 77
7.    Flaviviridae sohuld be in italics form (line 81)
8.    Were the females evaluated for other pathogenic infections such as syphilis, hepatitis B virus, hepatitis C virus and/or rubella and CMV?
9.    Please list at least the name of investigated genes in the 2.2. section
10.    The method depicted in gene expression graphs is 2^(-ΔΔct) Please revise the text in line 146
11.    No references have been included in lines 183-184
12.    When “higher”, “lower” more/les intense etc…are mentioned in the results section, P values should be included. When no difference are reported, p>0.05 should be included
13.    This sentence in lines 228-229 is unclear and should be rephrased
14.    In figure 3, no bars are showed below “***” or “**”, please include bars
15.    This sentence can be moved to the discussion (lines 310-312) as these results have already been described in lines 259-266
16. An additional limitation that should be stated is the relatively limited sample size. A higher number of available samples could increase the statistical power

Author Response

Reply to the Review Report (Reviewer 2)

The work is fine and well conducted. The authors investigated the impact of maternal overweight/obesity on the antiviral factors’ expression in the placental tissue of Zika-infected mothers. Results indicate that obesity associated with ZIKV infection leads to an impaired type I IFN downstream signaling pathway in the maternal–fetal interface.

We thank the reviewer for all the observations. We made the changes and added what was pointed out. We have addressed the comments bellow, as the modifications in the manuscript can be tracked in color yellow.

  1. Obesity rates among pregnant females worldwide and specifically in brasil should be included in the introduction (PMID: 30092099 and PMID: 34258876)

We appreciate for pointing this out and have added these references in the introduction (lines 60-62).

  1. This sentence in lines 56-60 is unclear and should be rephrased or subdivided in two sentences

We have now divided into two sentences, and hopefully it is clearer.

  1. Figures 1, 2 and 3 words should be enlarged as being difficult to read

We have increased the font size in all figures.

  1. Tables 1, 2 and 3 are actually figures. Please replace these figures with text

We have changed the Table I and II as Figures 1 and 3, as suggestions. However, Table 3 is more difficult to transform the histopathological changes in graphics, we maintained as Table format. 

  1. Pregnant females might become prone to viral infection as a consequence of the physiological immune modulation process occurring during the gestation period (PMID: 34970247, PMID: 33133091). Please include this notion

We appreciate for pointing this out and have discussed and added both this references in the discussion (lines 340-345).

  1. Please include a supporting reference for the statement present in line 77

The epidemiological reference for this sentence is now included.

  1. Flaviviridae sohuld be in italics form (line 81)

We apologize, and it is correct in the text now.

  1. Were the females evaluated for other pathogenic infections such as syphilis, hepatitis B virus, hepatitis C virus and/or rubella and CMV?

Yes, we used the clinical data from the hospitals to exclude placentas from women positive for TORCH (toxoplasmosis, rubella, cytomegalovirus, herpes, and other agents, including HIV, hepatitis B and C virus and syphilis). For the ZIKV’s cohort, we also excluded mothers that had other arboviruses, such as Dengue virus, and Chikungunya virus. This is described in the Human subject’s topic in the material and methods section.

  1. Please list at least the name of investigated genes in the 2.2. section

Yes, we have listed the names of the genes in the 2.2. section (line 148). The target genes were listed in the Supplementary Table 1,

  1. The method depicted in gene expression graphs is 2^(-ΔΔct) Please revise the text in line 146

The text that references Livak 2001 is correct. As we are working with samples not stimulated, we used as analysis the variation of 2^(- ΔΔct) method, that is 2^(- Δct), described in this literature, using the gene YWHAZ as an internal control.

  1. No references have been included in lines 183-184

For the statical analyses, we used the non-parametric and non-paired test between the groups, Mann-whitney.

  1. When “higher”, “lower” more/les intense etc…are mentioned in the results section, P values should be included. When no difference are reported, p>0.05 should be included.

Thank you for the suggestion. We have mentioned in the text as recommended.

  1. This sentence in lines 228-229 is unclear and should be rephrased

We have rephrased this paragraph, and hopefully it is clearer now (lines 238-244)

  1. In figure 3, no bars are showed below “***” or “**”, please include bars

We apologize for the inconvenience. It is probably due to the configuration problem, now we have changed the Figure.

  1. This sentence can be moved to the discussion (lines 310-312) as these results have already been described in lines 259-266

We removed this sentence, and a summary of our findings are presented in the discussion section (lines 323-326).

  1. An additional limitation that should be stated is the relatively limited sample size. A higher number of available samples could increase the statistical power

We agree with the reviewer, the sample size is an additional limitation. Unfortunately, to increase the accuracy of our cohort, we had to exclude samples with pre and post term gestational age, as well as other infections and obstetric complications. For this reason, and because it is a 2015/2016 cohort, we only had access to this sample number. We have included a sentence in the discussion (lines 357-360) pointing out this limitation.

Round 2

Reviewer 1 Report

The limitations of the study are explained.